# Association of Novel Advanced Glycation End-Product (AGE10) with Complications of Diabetes as Measured by Enzyme-Linked Immunosorbent Assay

**DOI:** 10.3390/jcm10194499

**Published:** 2021-09-29

**Authors:** Agnieszka Bronowicka-Szydełko, Małgorzata Krzystek-Korpacka, Małgorzata Gacka, Jadwiga Pietkiewicz, Urszula Jakobsche-Policht, Andrzej Gamian

**Affiliations:** 1Department of Biochemistry and Immunochemistry, Wroclaw Medical University, 50-368 Wroclaw, Poland; malgorzata.krzystek-korpacka@umed.wroc.pl (M.K.-K.); jadwiga.pietkiewicz@umed.wroc.pl (J.P.); 2Department of Angiology, Diabetes and Hypertension, Wroclaw Medical University, 50-556 Wroclaw, Poland; magacka@poczta.onet.pl (M.G.); urszula.jakobsche-policht@umed.wroc.pl (U.J.-P.); 3Laboratory of Medical Microbiology, Institute of Immunology and Experimental Therapy, Polish Academy of Sciences, 53-114 Wroclaw, Poland; andrzej.gamian@hirszfeld.pl

**Keywords:** advanced glycation end-products, diabetes, microangiopathy, glomerular filtration

## Abstract

Advanced glycation end-products (AGEs) contribute to vascular complications and organ damage in diabetes. The unique AGE epitope (AGE10) has recently been identified in human serum using synthetic melibiose-derived AGE (MAGE). We aimed at developing ELISA for AGE10 quantification, determining whether AGE10 is present in diabetic patients (*n* = 82), and evaluating its association with diabetic complications. In a competitive ELISA developed, the reaction of synthetic MAGE with anti-MAGE was inhibited by physiological AGE10 present in serum. In this assay, new murine IgE anti-MAGE monoclonal antibodies, which do not recognize conventional AGEs, a synthetic MAGE used to coat the plate, and LMW-MAGE (low molecular mass MAGE) necessary to plot a standard curve were used. AGE10 was significantly higher in patients with microangiopathy, in whom it depended on treatment, being lower in patients treated with aspirin. AGE10 levels were positively correlated with estimated glomerular filtration rate (eGFR) and negatively with creatinine. As a marker of stage ≥3 chronic kidney disease or microangiopathy, AGE10 displayed moderate overall accuracy (respectively, 69% and 71%) and good sensitivity (82.6% and 83.3%) but poor specificity (58.1% and 57.8%). In conclusion, newly developed immunoassay allows for AGE10 quantification. AGE10 elevation is associated with microangiopathy while its decrease accompanies stage ≥3 chronic kidney disease.

## 1. Introduction

Biochemical pathways contributing to diabetes-related damage of organs and blood vessels are being intensively investigated and the relevance of protein glycation is well recognized now. Glycation is a physiological process involved in aging, which, however, accelerates substantially in metabolic disorders [1,2]. The important role of protein glycation in conditions associated with hyperglycemia has been emphasized for many years. Most of advanced glycation end-products (AGE) are formed during the non-enzymatic Maillard reaction [3], wherein, Schiff base is formed in the initial stages. Then, it is converted into Amadori product which, in a cascade of various reactions, ultimately leads to the AGE formation. Glycation occurs between the reducing sugars or low molecular mass aldehydes (e.g., α-oxoaldehydes, hydroxyaldehydes) and the basic groups of proteins, lipids or nucleic acids, yielding compounds varying in structure and stability [4].

Long-lived proteins such as plasminogen activator inhibitor (PAI)-1, fibrinogen or albumin and extracellular matrix (ECM) proteins are particularly susceptible to glycation and subsequent formation of AGE [5,6]. In turn, glycation of proteins with short half-lives leads to the formation of Amadori products such as HbA1c in case of hemoglobin. Glycation impairs protein function to varying degrees. Moreover, modified proteins become resistant to degradation and are targeted by immune cells inducing inflammation [7], which, unresolved, may lead to cancer [8]. The accumulation of AGE has been shown in adipose tissue, muscles, nerves or blood plasma [9,10,11,12,13]. Moreover, the AGE presence may disturb the nerve impulse conduction in muscles contributing to amyotrophic lateral sclerosis called Charcot disease [14]. Increased AGE accumulation in tissues, as well as glucose intolerance or abnormal glucose metabolism are significant risk factors for accelerated atherosclerosis and cardiovascular diseases [15]. Hyperglycemia can lead to dysfunction of contractile myocytes (diabetic cardiomyopathy), occurring most often in patients with type 1 diabetes [14,16]. AGEs may damage ryanodine receptor in myocytes altering calcium transfer from sarcoplasmic reticulum to mitochondria in the senescent myocardium [17]. The cardiovascular system diseases, especially coronary heart disease (CHD), are the leading cause of death in patients with type 2 diabetes [15]. The progressive glycation can also affect the central nervous system (CNS), damaging the brain and peripheral nerves and autonomic nervous system (ANS) [18,19].

Kidneys are among organs particularly affected by AGE accumulation [20]. The loss of negative charge of the nephron filtration barrier accelerates the process, causing oxidative stress and upregulating expression of pro-inflammatory cytokines [20]. Animal studies have shown that AGE exacerbated diabetic nephropathy, which, in turn, contributed to the thickening and hardening of the glomerular basement membrane, fibrosis of tubular structures of renal mesangial cells, and increased the expression of TGF-β and collagen IV [21]. Moreover, as a result of chronic hyperglycemia, the synthesis of angiotensin II increases, rising the blood pressure, aggravating inflammation and oxidative stress, and leading to proteinuria and, ultimately, glomerular fibrosis [22].

Recently, we have characterized a new synthetic glycation end-product, MAGE, which is produced by the modification of proteins by melibiose. MAGE is a synthetic analogue of AGE epitope that presence has been detected in tissues from invertebrates, such as snails, and vertebrates, including fish, frog, chicken, pig, horse, rat, rabbit and human. MAGE creates isomers of fructosamine that contain an attached disaccharide, in which both carbohydrate moieties (i.e., galactose and glucose) are in a closed form, in contrast to fructosamine formed from glucose or fructose [23]. The exact structure of AGE epitope remains unraveled while that of MAGE is elaborated in Staniszewska et al. [24]. Importantly, autoantibodies against this epitope have been detected in serum samples from diabetic patients [24]. They had high specificity because they did not react with proteins glycated by different glycation agents such as glucose, fructose, methylglyoxal, lactose, trans-2-nonenal, glycoaldehyde or maltose. This AGE epitope, according to the AGE’s nomenclature [25,26], was named AGE10.

The clinical significance of AGE10 has not been established yet as there was no assay developed allowing for its quantification. Therefore, the aim of present study was to develop an ELISA assay allowing for AGE10 quantification in human serum using monoclonal anti-MAGE antibodies generated in mice. The devised method was then validated on clinical samples and AGE10 association with biochemical parameters and clinical presentation of diabetic patients was determined.

## 2. Materials and Methods

### 2.1. Analytical Methods Used during Developing an Immunoassay for AGE10 Quantification

An overview of a process of developing competitive immunoassay (ELISA) for AGE 10 detection is presented in Figure 1.

#### 2.1.1. Preparation of ELISA Reagents (Step 1)

Development of competitive ELISA required synthesis of antigens for plate-coating (Step 1.1 in Figure 1), synthesis of low molecular mass MAGE as standards (LMW-MAGE) for a standard curve (Step 1.2 in Figure 1), and synthesis of primary antibodies (Step 1.3 in Figure 1).

Preparation of plate coating-antigen MAGE

Preparation of plate coating-antigen MAGE (Step 1) included synthesis of glycation products of reaction of myoglobin with melibiose (MB-MELs), their separation, and identification of fraction (MAGE) the most reactive against anti-MAGE antibodies (Figure 1).

MB-MELs synthesis (Step 1.1.1)

Appropriate amount of myoglobin (MB) from Merck (Darmstadt, Germany) was mixed with melibiose (MEL) (Merck), dissolved in 1 mL H_2_O miliQ, frozen at −80 °C, lyophilized and placed for 45 min at 85 °C in a microwave reactor (Initiator 2.5, Biotage, Uppsala, Sweden), equipped with a pressure compressor (Jun-Air Model 2xOF302-40B 230 V/50 Hz, Redditch, UK) at constant power of 200 W.

MB-MELs separation—FPLC and SDS/PAGE (Step 1.1.2)

Samples after synthesis were treated with 1 mL of 0.01 M ammonium acetate buffer pH 6.8 and subjected to chromatographic separation. Chromatography was carried out in 10 mM ammonium acetate buffer, pH 6.8, on a column (1.6 cm × 100 cm, XK 16/100, Pharmacia, Apeldoorn, the Netherlands) packed with hydroxymethyl resin (Toyopearl resin HW-55S, Tosoh, Thermo Fisher Scientific, Massachusetts, GA, USA), equilibrated in advance with the same buffer. The separation was performed in the fast protein liquid chromatography (FPLC) system (ÄKTAexplorer, Amarsham, Pharmacia Biotech, Ramsey, NJ, USA), recording the elution profile at three wavelengths: 225 nm, 280 nm and 297 nm. The separation was carried out at the flow rate of 0.1 mL/min and the volume of the collected fractions was 1 mL. Peak fractions were combined, lyophilized and weighed.

Chromatographically-separated modified proteins were subsequently analyzed by SDS/PAGE electrophoresis on a 12% polyacrylamide gel. Protein samples (10 μg) were suspended in sample buffer (0.06 M Tris-HCl pH 6.8, 2% SDS, 10% glycerol, 0.025% bromophenol blue, 5% β-mercaptoethanol) and applied on the gel. Electrophoresis was performed at a constant current of 40 mA for 1.5 h using Mini-Protean Tetra Cell electrophoresis apparatus (Bio-Rad, Hercules, CA, USA). A 0.25% Coomassie Brilliant Blue R-250 solution (Bio-Rad) was used for protein staining and an aqueous solution of 50 mM methanol and 75 mM acetic acid was used to decolorize background. The results were analyzed using Vilber Lourmat gel/blot analysis system (Transilluminator UV/white light Z363820, Merck, Darmstadt, Germany).

MAGE identification—ELISA (Step 1.1.3)

The plates (Nunc MaxiSorp^®^, Thermo Fisher Scientific, Massachusetts, GA, USA) were coated with MB-MEL products contained in the fractions obtained in the FPLC chromatographic separation (100 μL/well) and incubated for 5 h at room temperature, followed by an overnight incubation at 4 °C. Subsequently, plates were washed 3-times (3 × 400 µL/well) with Tris-Buffered Saline, 0.1% Tween 20 Detergent (Merck) TBS-T and blocked for two hours at room temperature with 5% defatted milk dissolved in TBS. After washing, plates were treated with anti-MAGE (100 μL/well; 1:2000) for one hour at 37 °C, washed, and incubated for 2 h at room temperature with horseradish peroxidase (HRP)-conjugated goat anti-mouse-IgE (Jackson ImmunoResearch Laboratory, West Grove, PA, USA) (1:3000). Following washing, plates were incubated with o-phenylenediamine dihydrochloride (OPD; Merck) for 10 min at room temperature. Absorbance was measured at 492 nm with microplate reader (LabEnspire, PerkinElmer, Waltham, MA, USA).

Preparation of standards—LMW-MAGE (Step 1.2)

Preparation of LMW-MAGE as standards included four steps: synthesis of low molecular mass glycation products from N-acetyl-α-lysine and melibiose (NAc_α_Lys-MEL), their separation followed by the selection of optimal LMW-MAGE, based on their inhibitory potential, and the determination of molecular mass of selected LMW-MAGE.

NAc_α_Lys-MEL synthesis (Step 1.2.1)

The mixture of N-acetyl-α-lysine (Merck) and melibiose (Merck, Darmstadt, Germany) was frozen at −80 °C and freeze-dried. The reaction was then carried out in a microwave reactor (Initiator 2.5, Biotage) with a pressure compressor (Jun-Air) Glycation was performed at 60 °C for 25 min at a constant power of 200 W.

NAc_α_Lys-MEL separation—FPLC (Step 1.2.2)

Mixture of glycation products was treated with 1 mL of 0.01 M ammonium acetate buffer pH 6.8 and subjected to chromatographic separation on 1.6 cm × 100 cm, XK 16/100 column (Pharmacia), packed with hydroxymethyl resin (Toyopearl resin HW-40S, Tosoh, Thermo Fisher Scientific). The separation was performed in the FPLC system (Pharmacia), at the flow rate: 0.1 mL/min, and the volume of the collected fractions was 1 mL. The elution profiles were measured at 225 nm, 280 nm and 297 nm. The obtained fractions were combined, lyophilized and weighed. Samples were then desalinated on Bio-Gel P-2 (Bio-Rad) on XK16/100 column (Pharmacia) in H_2_OmiliQ at a flow rate of 0.1 mL/min. The volume of the collected fractions was 1 mL. The elution profile was determined at 225 nm, 297 nm and 325 nm. After separation, the fractions were combined, frozen, lyophilized and weighed.

LMW-MAGE selection—competitive ELISA (Step 1.2.3)

ELISA plates (Nunc MaxiSorp^®^, Thermo Fisher Scientific) were coated with 1 μg/well synthetic medium-cross-linked MAGE (FPLC-separated), dissolved in 100 μL of 120 mM carbonate buffer pH 9.6 (20 mM sodium carbonate, 100 mM sodium bicarbonate). Plate incubation, blocking and washing were performed as described above for sandwich ELISA. Samples containing 0, 5, 10, 20 and 40 µg LMW glycation products in 160 µL of PBS were mixed 1:1 with anti-MAGE antibodies (diluted 1:2000 in PBS), incubated (1 h at 37 °C) and applied on a plate in triplicates (100 µL/well). Following 2 h incubation at 37 °C and an overnight incubation at 4 °C, plates were washed 3 times with TBS-T buffer. Subsequently, 100 µL of secondary murine anti-IgE antibodies conjugated with HRP (Jackson ImmunoResearch), diluted 1:4000 in PBS, were added and incubated at room temperature for 2.5 h. Following washing, plates were incubated with OPD (10 min at room temperature). Absorbance was measured at 492 nm with microplate reader (PerkinElmer).

Determination of LMW-MAGE mass (Step 1.2.4)

Molecular masses of LMW-MAGE products were determined by liquid chromatography mass spectrometry (LC-MS) as described earlier [23]. Samples were separated on the nanoAcquity UPLC system (Waters, Milford, MA, USA) under the conditions described in Appendix A. The total separation time (including the time for column regeneration) was 30 min.

A Symmetry C18 chromatography column (100 µm × 100 mm; grain diameter: 3.5 µm) was used in the chromatography. The mobile phase was a mixture of water (A) and acetonitrile (B). Before loading the samples on the analytical column, the material was prebound to the pre-column and rinsed for 5 min with a mixture of water (99.5%) and acetonitrile (0.5%), at a flow of 5 µL/min, for additional purification from possible contamination. The analytical column was thermostated and its temperature was 35 °C. The samples were stored in an autosampler at 7 °C. The concentration of compounds in the samples was approximately 0.1 µg/µL. The volume of 5 µL of LMW-MAGE products from the given sample were injected onto the column. Spectrometric analyzes were performed on a hybrid mass spectrometer (XevoG2 Q-TOF, Waters) equipped with a nanoelectrospray source under the conditions described in Appendix A.

Measurements were performed in the high sensitivity mode of the analyzer, under positive polarization conditions. The reading was carried out for 20 min for ions in the mass range: (initially) 300 to 1500 *m*/*z*, and (in later analyzes) 80 to 800 *m*/*z*. During the measurement, the spectra were recorded with the frequency of 2/s in the “centroid” mode.

#### 2.1.2. Development AGE10 ELISA (Step 1.2)

Preparation of serum samples from patients and standards (LMW-MAGE)

In the final ELISA, plates were coated with MAGE (antigen) and serum samples from patients were source of AGE10 epitopes. As they might be present either on protein surface or buried within, thus unavailable for antibodies, serum samples were subjected to proteinase K treatment. Serum samples (35 μL) were suspended in 70 μL of 60 mM Tris- HCl pH 8.0 with 5 mM CaCl_2_ and 50% glycerol, containing proteinase K (30 U/mg) from Tritirachium album (Merck) at 0.2 mg/mL. Samples were mixed, centrifuged (2000× *g*, 1 min) and incubated at 50 °C overnight. Then, proteinase K was denatured (110 °C, 15 min) and samples were cooled and centrifuged (15,000× *g*, 15 min). Aliquots of 90 μL were diluted in PBS (1:1) (Merck) devoid of Ca^2+^ and Mg^2+^, mixed (1:1, *v*/*v*) with anti-MAGE in PBS (final dilution 1:3000) and subsequently incubated at 37 °C for 1 h.

Standards—LMW MAGE in PBS at concentration range 0-400 μg/mL—were mixed (1:1, *v*/*v*) with anti-MAGE in PBS (final dilution 1:3000) and subsequently incubated at 37 °C for one hour.

AGE10 ELISA

Microtiter plates were coated with MAGE (1 μg/well; 5 h at room temperature), washed with TBS-T and blocked overnight with non-fat milk at 4 °C. Following washing, serum samples and standards with murine anti-MAGE monoclonal antibodies (1:2000), prepared as above, were applied and incubated at 37 °C for 2 h. After washing, goat anti-mouse IgE antibodies conjugated with HRP (1:3000) (Jackson ImmunoResearch) were applied for 2.5 h (room temperature) and then washed and OPD was added (10 min, room temperature). Absorbance was read at 492 nm on microplate reader (PerkinElmer). All samples and standards were assessed in triplicates.

### 2.2. Study Population

Serum samples from 82 patients (51 women and 31 men) treated in the Department of Angiology, Hypertension and Diabetes of Medical University of Wroclaw from May 2012 to November 2012 were collected. The research was approved by the Bioethics Committee of the Medical University in Wrocław (No. KB-384/2012).

Laboratory parameters including data on glycated hemoglobin (HbA1c), glucose, C-reactive protein (CRP), total cholesterol, LDL, HDL, triglycerides, uric acid, albuminuria, and creatinine concentration were quantified using routine diagnostic procedures and collected prospectively. Glomerular filtration rate (eGFR) was calculated using the MDRD formula. Diabetic complications were defined as follows: macroangiopathy as a presence of peripheral arterial disease and/or heart disease without myocardial infarct and/or myocardial infarct in the past and/or ischemic stroke in the past on the basis of medical documentation; peripheral arterial disease as an ankle brachial index less than 0.9 and/or revascularization in anamnesis (percutaneous angioplasty or traditional surgical treatment) and/or significant stenosis of arteries in legs (more than 70%); heart disease as a revascularization without myocardial infarct performed in the past (percutaneous angioplasty of coronary arteries/coronary artery bypass grafting) or a positive exercise stress test in the anamnesis; myocardial infarct as typical changes in the past, including elevated cardiac biomarkers and changes on an electrocardiogram (ST segment changes, new left bundle branch block, or pathologic Q waves); ischemic stroke as typical clinical symptoms and ischemic changes in computed tomography or in magnetic resonance imaging in the past; microangiopathy as a presence of retinopathy or nephropathy on the basis of medical documentation; retinopathy as typical changes for diabetes in retina in ophthalmological examination and/or in fluorescein angiography; polyneuropathy as typical for diabetes changes in peripheral nerves in neurological examination and/or result of electromyography; nephropathy as microalbuminuria defined as urinary albumin excretion in the range of 30–299 mg/24 h or macroalbuminuria defined as urinary albumin excretion ≥300 mg/24 h; hyperlipidemia as plasma concentrations of total cholesterol ≥175 mg/dL, triglyceride ≥150 mg/dL and HDL level <40 mg/dL in men or HDL <45 mg/dL in women or the level of LDL ≥100 mg/dL (or above the recommended values for a given cardiovascular risk group); hypertension as systolic blood pressure ≥140 mmHg and/or diastolic blood pressure ≥90 in two measurements or treatment with antihypertensive drugs. The degree of damage in the kidney disease was estimated by serum creatinine/estimated glomerular filtration rate (eGFR) with eGFR is less than 90mL/min/1.73m^2^ considered abnormal.

Blood (30 mL) was collected from the median cubital vein into plastic tubes with a clot activator (Becton Dickinson Vacutainer, East Rutherford, NJ, USA). The blood was collected in the morning and stored for up to three hours at room temperature. Then, samples were centrifuged (2000× *g* for 15 min) and the supernatant was aliquoted and frozen at −20 °C until analysis.

#### Preparation of Murine Monoclonal Anti-MAGE Antibodies (Step 1.3)

Groups of 6-week-old male BALB/c mice were injected with a mixture of 50 µg of rabbit glycated immunoglobulins (RIg-MAGE) and 50 µg horse glycated myoglobin (MB-MAGE) emulsified in complete Freund’s adjuvant (CFA). The first dose was given subcutaneously near the inguinal lymph nodes. After one month of a single dose injection, mice were immunized intraperitoneally with the same dose of antigens mixture, ten times with 2-week intervals. Blood samplings for detection of specific antibody production were performed under anesthesia of animals. Fusions of myeloma cells with immune spleen cells were carried out using the method of Kohler and Milstein procedure modified by Dippold [27]. Cells were fused with SP2/0 mouse plasmacytoma cells for one-minute incubation in 0.2 M of polyethylene glycol 1500 solution (BDH, Laboratory Supplies, UK) prepared by dissolving 5 g of autoclaved PEG 1500 in 7.5 mL of PBS, pH 7.4 containing 15% of dimethyl sulfoxide. Experiments were approved by the Local Animal Care and Use Committee at the Hirszfeld Institute of Immunology and Experimental Therapy PAS (LKE 53/2009). The class of anti-MAGE were determined by Rapid Mouse Isotyping Kit-Gold Series, LFM-ISO-1-5 (RayBiotech Inc., Norcross, GA, USA).

The presence of clones producing anti-MAGE antibodies were detected by ELISA. Wells of a plate were coated by BSA-MAGE (bovine serum albumin–MAGE), MB-MAGE (myoglobin–MAGE) and Lys-MAGE (lysozyme–MAGE) and their unglycated equivalents (BSA, MB, Lys). Nunc Maxisorp plates were coated with an appropriate antigen (0.5 µg/well) dissolved in 100 µL of carbonate buffer pH 9.6 (16 mM sodium carbonate, 34 mM sodium bicarbonate) and incubated overnight at 4 °C. The next day, the plate was washed 3-times with TBS with 0.05% Tween 20 (TBS-T), pH 7.4 (15 mM Tris, 150 mM NaCl, 0.05% Tween) and blocked overnight at 4 °C with 5% ovalbumin (Sigma-Aldrich, St. Louis, MO, USA) in TBS at 200 µL/well. The next day plate was washed as before. Then, 50 µL medium containing hybridoma clones (growth after cell fusion) was added into each well in triplicates and incubated at 4 °C overnight. The next day the plate was washed as before and 1:4000 diluted HRP-conjugated goat anti-mouse IgE (Jackson ImmunoResearch Laboratory) in TBS was added to each well (at 50 µL/well) and incubated at room temperature for 2.5 h. Then the plate was washed as before and the reaction was caused by solution of 30 mg OPD (Thermo Scientific) dissolved in 10 mL of citrate buffer, pH 4.5, containing 50 mM citric acid, 70 mM sodium citrate, 5% methanol and 0.03% H_2_O_2_ (at 50 µL/well). The plate was incubated for 10 min at room temperature and then the reaction was stopped by 40% H_2_SO_4_. Absorbance was measured at 492 nm in microplate readers (Microplate Spectrophotometer, Powerwave XS, Biotek, Milan, Italy).

### 2.3. Statistical Analysis

The normality of distribution was tested using D’Agostino-Pearson test and homogeneity of variances using Levene test. Data on AGE10 were normally distributed and are therefore presented as means with 95% confidence interval (CI). They were analyzed using one-way ANOVA with Tukey–Kramer post-hoc test (multigroup comparisons) or *t*-test for independent samples (two-group comparisons). Two-way ANOVA or analysis of covariance (ANCOVA) were used to co-examine data such as microangiopathy and treatment or microangiopathy and eGFR effects on AGE10. Correlation analysis was conducted using Pearson correlation test. The discriminative power of AGE10 as a biomarker was tested using receiver operating characteristics (ROC) curve analysis.

All calculated probabilities were two-tailed and p-values ≤0.05 were considered statistically significant. The analyses were performed using MedCalc^®^ Statistical Software version 19.6 (MedCalc Software Ltd., Ostend, Belgium; https://www.medcalc.org; accessed on 3 September 2020).

## 3. Results

### 3.1. ELISA Reagents

#### 3.1.1. Plate-Coating Antigen (MAGE)

To enable quantitative determination of AGE10 in biological material, an ELISA assay using anti-MAGE antibodies was developed. In this assay, the wells of plates were covered with synthetic MAGE, prepared by glycation of myoglobin with melibiose in a microwave reactor. Reaction products were then separated by FPLC chromatography yielding three peaks—highly cross-linked glycation products denoted 1(A), moderately cross-linked glycation products denoted 1(B), and lowly cross-linked glycation products denoted 1(C), which were subsequently analyzed in SDS-PAGE. Typical chromatogram and gel image are presented in Figure 2a,b, respectively. The moderately cross-linked fractions, 1(B), were then found to possess the highest reactivity towards anti-MAGE antibodies as detected in ELISA (Figure 2c).

#### 3.1.2. Standards (LMW-MAGE)

The mixture of NAc_α_Lys-MEL products obtained during the microwave synthesis was subjected to chromatographic separation on a HW-40S column. The elution profiles (Figure 3a) yielded two peaks, denoted 1(A) and 1(B). To desalinate samples, fractions constituting a 1(B) peak were subjected to FPLC chromatographic separation on a Bio-Gel P-2 column, resulting in further separation into three peaks, denoted 1(B0), 1(B1) and 1(B2) (Figure 3b). Testing resulting peaks for LMW-MAGE presence using competitive ELISA showed 1(B1) peak to be the most efficient in inhibiting MAGE/anti-MAGE reaction and therefore to contain an analogue of AGE10 epitope (Figure 3c). All peaks were subjected to mass-spectrometry analysis in order to determine the molecular mass of respective LMW glycation products (Figure 3d). The presence of LMW-MAGE, with a mass to charge (*m*/*z*) ratio of 513.23, appearing at approx. 2.1 s, was detected mainly in 1(A) and 1(B1) samples—the peaks showing the highest inhibitory activity against MAGE/anti-MAGE reaction.

#### 3.1.3. Preparation of Murine Anti-MAGE Monoclonal Antibodies

Selection of clones producing monoclonal anti-MAGE antibodies was performed with ELISA. Three of 132 clones (Nos. 10, 19, 49) produced antibodies reacting with BSA-MAGE, MB-MAGE, LYS-MAGE (Figure 4). The reactivity of antibodies with BSA, MB or Lys was the screening control, respectively.

The largest and statistically significant difference in reactivity between produced monoclonal anti-MAGE antibodies using BSA-MAGE, MB-MAGE and LYS-MAGE and corresponding unmodified proteins: BSA, LYS, MB was observed for Clone Number 10. Although antibodies present in Clones No. 19 and 49 were reactive with BSA-MAGE, MB-MAGE and LYS-MAGE, they were also very reactive with unmodified proteins (BSA, MB, LYS) and these differences were not statistically significant. The appropriate ELISA showed that the anti-MAGE antibodies were of IgE class.

### 3.2. AGE10 ELISA

The overall scheme of developed AGE10 ELISA is presented in Figure 5.

In order to optimize the test, it was carried out under various conditions of time and temperature and at varying dilutions of reagents. Serum dilution was tested within 6–96 times range and 12-fold dilution was found optimal. Anti-MAGE dilutions tested were within 50–10,000 times range and those of anti-mouse IgE antibodies within 100–10,000 times range and the dilutions of 3000 times was found optimal on both accounts. Mass of MAGE applied into wells within 0.1–20 µg/well range was assessed and 1 μg was found optimal. Optimal incubation time with primary and secondary antibodies was 3 h for primary and 2 h for secondary antibodies, chosen from interval of 0.5–4 h tested. Regarding incubation temperature, test performed better if incubation with both primary and secondary antibodies was conducted at 37 °C than 22 °C (room temperature) with an incubation with primary antibodies followed by overnight incubation at 4 °C. In addition, the reactivity of the antibodies against the native myoglobin and the non-microwave treated MB/MEL mixture was also evaluated and none was found.

Intra- and inter-assay coefficients of variation of the method (CV%) were deter-mined to be 5 ± 0.2% and 5 ± 0.4%, respectively (*n* = 6). Limit of quantification (LOQ) for developed assay was 18 µg/mL and limit of detection (LOD) was 5.8 µg/mL.

### 3.3. Determination of AGE10 Concentration in Sera—Validation of AGE10 ELISA

The characteristics of the studied population are shown in Table 1.

The utility of developed AGE10 ELISA was assessed using serum samples from patients with type 2 diabetes mellitus. In the tested cohort, AGE10 concentrations ranged from 0 to 400 μg/mL. AGE10 concentrations below limit of detection were observed in 13 patients (15.8%).

The association between AGE10 concentration and diabetes complications showed AGE10 to be significantly higher in patients with microangiopathy (Table 2).

The possible relationship between AGE10 and indices of inflammation (CRP), kidney function (albuminuria/creatinine index in morning urine sample and eGFR), lipid metabolism (HDL, LDL, triglycerides), glycemia and its control (glucose and HbA1c concentration) was investigated (Table 3). AGE10 was inversely correlated with creatinine concentration and positively with eGFR. As kidney function and microangiopathy are related, both parameters were co-examined using analysis of covariance (ANCOVA) and found to have an independent effect on AGE10 (*p* = 0.05 for microangiopathy and *p* = 0.004 for eGFR). These two factors explained 17% of variance in AGE10 concentration (R^2^ = 0.168) observed in patients with type 2 diabetes.

None of applied medication had significant impact on AGE10 concentration in diabetic patients in general (Table 4). However, aspirin treatment seems to abolish the difference in AGE10 concentration between patients without and with microangiopathy (Figure 6).

### 3.4. AGE10 Potential as a Biomarker

The potential of AGE-10 as a biomarker of microangiopathy and stage ≥3 chronic kidney disease, defined as eGFR <60, was assessed.

As a marker of microangiopathy presence, AGE10 had 71% overall accuracy (Figure 7a). At optimal cut-off of >83.4 μg/mL, associated sensitivity was 83.3% and specificity of 57.8%. As a marker of stage ≥3 chronic kidney disease, AGE10 had 69% overall accuracy (Figure 7b). At optimal cut-off of ≤133 μg/mL, associated sensitivity was 82.6% and specificity of 58.1%.

## 4. Discussion

Non-enzymatic modification of macromolecules by sugars is a main culprit in the pathogenesis of diabetic complications [28,29,30] and the resulting products may serve as biomarkers of metabolic disorders and treatment efficacy. Currently, fructosyllysine, glycated albumin (GA) and glycated hemoglobin (HbA1c) are used in clinical practice. However, they are not proper AGE but less stable Amadori products [31,32,33]. Therefore, longer-lived and stable AGE markers are sought after. Immunoassays, due to their relative simplicity and low cost, are frequently used as diagnostic tools. Regarding AGE, however, their widespread use is hampered by lack of assays allowing to quantify individual AGE epitope [34,35]. Except for the most known AGE—carboxymethyllysine (CML), commercially available immunoassays detect a mixture of various AGEs. Still, a body of evidence is gathering showing that individual AGEs have indeed potential as markers of diabetic complications, e.g., a methylglyoxal-derived hydroimidazolone-1 is proposed as an early marker of atherosclerosis in childhood diabetes [36], AGE4 (albumin modified by methylglyoxal) is an independent predictor of polyneuropathy in diabetes [35] and AGE1 (albumin modified by glucose) is significantly associated with lipid abnormalities in diabetes [37], justifying efforts put into developing epitope-specific immunoassays.

Here, we propose an immunoassay allowing for quantification of modified serum proteins using synthetic Melibiose derivative MAGE. Melibiose (α-D-gal-(1→6)-D-glc) enters the body with a plant-based diet, in particular with products such as cocoa beans and soy roots and stems [38,39] or with honey [40]. It can also be provided by gut microbiota such as Bifidobacterium breve [41]. This disaccharide crosses the intestinal wall using the passive paracellular permeation pathway [42] and might react with plasma and extracellular matrix proteins. MAGE is an analog of an adduct commonly found in several tissues of humans and various animal species [24]. That epitope recognized by anti-MAGE monoclonal antibody might be clinically relevant has been shown in sera from patients with diabetes and atherosclerosis [24].

A competitive assay developed here employs novel monoclonal anti-MAGE anti-bodies, synthetic MAGE, and LMW-MAGE used for preparing a standard curve. In order to obtain MAGE and LMW-MAGE, it was necessary to develop an in vitro synthesis method. Thus, so far, the most common approach to AGE synthesis consists of pro-longed (several weeks) incubation of a given glycating factor (predominantly glucose) with a model protein at a specific temperature and in specific buffers or other polar solvent, which subsequently necessitates time-consuming dialysis of obtained AGE [33,43]. Frequently, additional prior incubation of model protein with H2O2 is conducted as the oxidized protein is more susceptible to glycation [44]. Manipulating glycation conditions—incubation time, temperature, pH, concentration of glycating factors and concentration of model proteins—has been used in order to shorten the process and increase its efficiency. Recently, carrying out the process using ultrasound [43] or microwaves [45] has been successfully tried. The method of MAGE and LMW-MAGE synthesis using microwaves at anhydrous conditions presented here allowed to shorten the process to 40 and 25 min, respectively.

The developed ELISA enabled AGE10 quantification in patients with various diabetic complications. We showed that its level is higher in patients with microangiopathy. Diabetic microangiopathy is caused by damage to the small blood vessels, mainly the retinal capillaries, nerves and glomeruli. In retinopathy, AGEs accumulate in the retina, vitreous fluid and the lens of the eyes, impeding vision and light scatter [46]. AGEs also change the conformation of eye’s proteins reducing the transparency of the lens [47]. In addition, retinal pigment epithelial cells contain RAGE, AGE receptors, and their interaction maintains inflammation, increases production of vascular endothelial growth factor, causes neurodegeneration and microvascular disorders [48,49]. In neuropathy, AGEs cause the thickening of the basement membrane and increase parietal permeability. The AGE-RAGE interactions in the perineural and endoneural blood vessels lead to vascular malfunction and the initiation of hypoxia. Moreover, AGEs weaken the immune system, increasing the likelihood of skin wounds in the feet [50,51]. Likewise, AGEs in nephropathy are accumulated in kidneys within the glomerular basement membrane, podocytes, tubules, endothelial cells and mesangium [52] and react with RAGEs, exacerbating inflammation [53]. Moreover, AGEs activate parietal epithelial cells, leading to thickening of the Bowman’s capsule surrounding the glomerulus, altering the efficacy of renal filtration [35]. AGE4 causes mitochondrial dysfunction and stress in endoplasmic reticulum of nephrons [54]. Accumulation of AGEs results in renal toxicity which gradually reduces the kidney filtration and leads to chronic kidney disease [55]. Moreover, the level of fluorescent LMW-AGEs has the significant relation with mortality in patients receiving chronic hemodialysis [56]. The association between AGE10 and microangiopathy observed in current study was evident solely in patients non-treated with aspirin.

The AGE10-reducing effect of aspirin agrees with previously reported drug im-pact on AGE. Low-dose aspirin has been shown to reduce blood glucose level in animal models of diabetes [57], therefore, reducing the rate of glycation strongly dependent on concentration of glycating agents. Aspirin decreases AGE accumulated in tissues by blocking the renin-angiotensin system and reducing inflammation and generation of reactive oxygen species (ROS) in addition to maintaining glycemic control. Regarding ROS, aspirin [34] chelates transition metals thus prevent Fenton reaction and can scavenge free carbonyls. Moreover, aspirin prevents glycation of plasma proteins, including hemoglobin, but the molecular mechanism is not fully understood. Probably, aspirin acetylates free amino groups of a protein, although it is likely only one aspect of its anti-glycation activity [36]. An example of a glycation product, the formation of which is inhibited by aspirin, is pentosidine. The aspirin action contributes to reduction of the amount of pentosidine in collagen preventing retinopathy [37].

The inverse relationship between AGE10 and kidney function observed here, that is, a positive correlation with eGFR and negative with creatinine, might seem counter-intuitive as AGE10 accumulates in pathological conditions. However, it is speculated that accelerated AGE formation from plasma proteins might be a protective mechanism, allowing long-lived proteins, such as extracellular matrix proteins, to avoid glycoxidative damage. Enolase, a glycolytic enzyme may serve as an example. It can trap reactive dicarbonyls and form AGEs, making other proteins less susceptible to modification. The concentration of enolase in the cytosol of cells exceeds the level necessary for the course of glycolysis, and the excess of this protein is used by the cell for other, non-enzymatic purposes [58,59]. Still, the impairment of renal filtration is associated with proteinuria and therefore the excretion of AGE10 might be accelerated. As such, it would be of interest to determine AGE10 concentrations in urine of diabetic patients at various stages of kidney failure.

As an interest in AGEs as possible diagnostic markers is increasing, we tested the discriminative power of this new epitope, AGE10, as an indicator of microangiopathy and stage ≥3 chronic kidney disease. AGE10 had moderate overall accuracy, defined as area under ROC curve expressed in %. At optimal cut-off, AGE10 occurred to be satisfactorily sensitive marker but characterized by poor specificity, regardless, whether indicating microangiopathy or stage ≥3 chronic kidney disease.

Although AGEs are potential biomarkers of various metabolic disorders because they are diverse, persistent compounds, and their formation is intensified under oxidative and carbonyl stress, they are present in blood in trace amounts. AGE10 is one of the few AGEs for which a method of quantitative determination in biological material has been developed. Many of them have so far been impossible to detect. Due to the fact that the structure of the AGE10 epitope has not yet been known, it is not possible to determine AGE10 using more sensitive methods, e.g., LC-MS mass spectrometry. Still, our study has several limitations that ought to be mentioned. As the exact structure of AGE10 epitope is only under investigation, the assay is utilizing its synthetic analogue—MAGE—the structure of which has already been resolved [24]. Consequently, molecular mass of AGE10 remains unknown and its level has to ex-pressed in mass units. Regarding the devised assay, a cross-reactivity with other AGEs has not been tested in the present paper but specificity of anti-MAGE antibodies has previously been evaluated [24]. Anti-MAGE antibodies have shown no cross-reactivity with proteins modified with lactose, glucose, fructose, methylglyoxal or glyceraldehyde. Lastly, the developed ELISA is tested on non-homogeneous cohort of patients with respect to diabetes type, severity and duration which may affect study results presenting AGE10 association with diabetes complications.

## 5. Conclusions

Presented methods allowed for preparation of assay components (MAGE, LMW-MAGE, murine monoclonal IgE anti-MAGE antibodies), necessary for developing the competitive ELISA for AGE10 quantification. Applied in a cohort of diabetic patients, the developed assay demonstrated AGE10 elevation in association with microangiopathy while its decrease in stage ≥3 chronic kidney disease. Further studies are needed to resolve structure of AGE10 and clarify its clinical relevance.

## Figures and Tables

**Figure 1 jcm-10-04499-f001:**
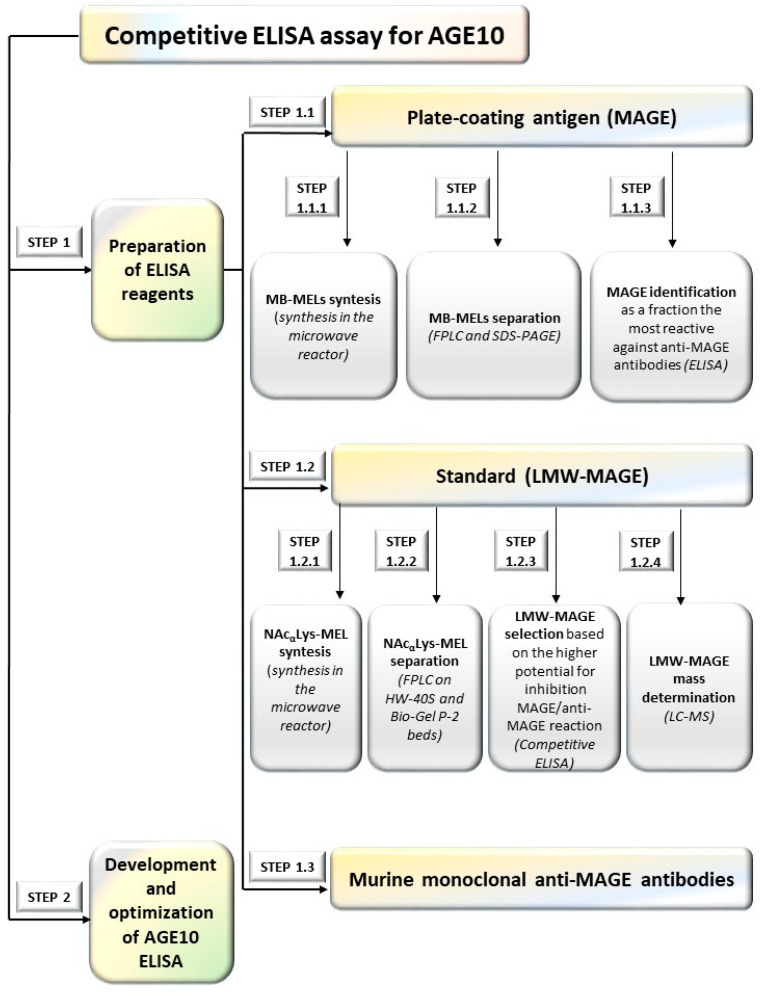
An overview of a process of AGE10 ELISA development; FPLC—the fast protein liquid chromatography, LC-MS—liquid chromatography mass spectrometer, LMW-MAGE—low molecular mass MAGE, MAGE—melibiose-derived AGE, MB-MELs—products of reaction of myoglobin with melibiose, NAc^α^Lys-MEL—products of reaction of N^α^-acetyllysine with melibiose.

**Figure 2 jcm-10-04499-f002:**
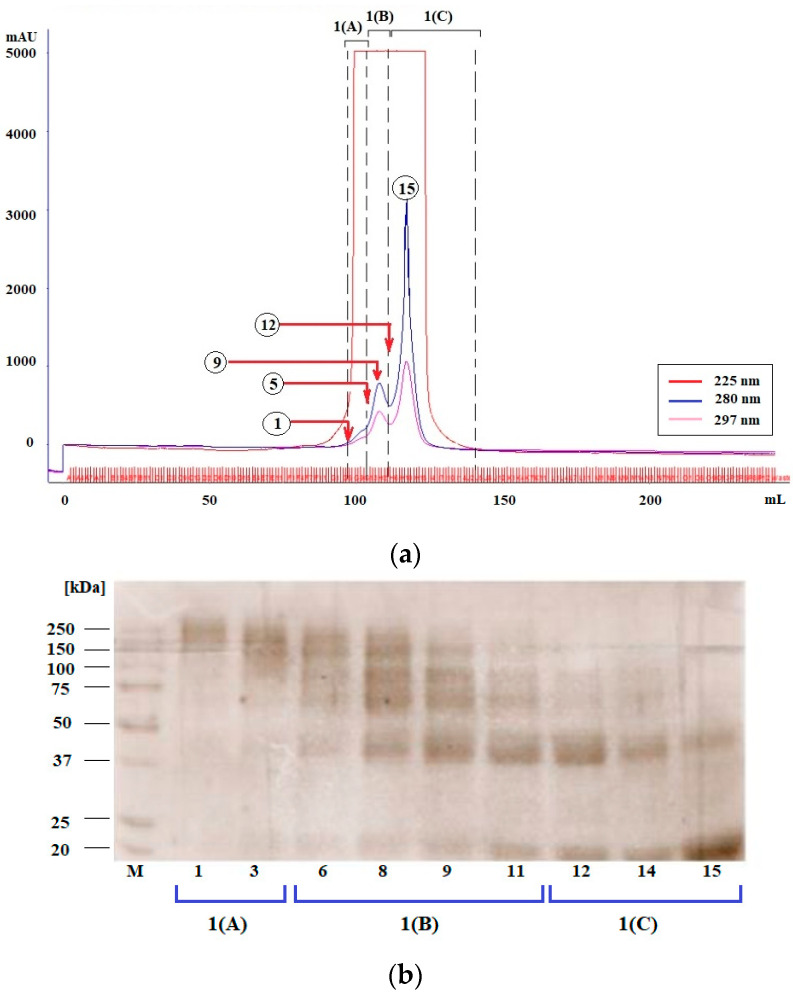
Glycation of myoglobin with melibiose and selection of MAGE with the highest reactivity towards anti-MAGE mouse monoclonal antibodies: (**a**) FPLC chromatogram with marked three peaks denoted 1(A), 1(B), and 1(C); (**b**) SDS-PAGE image of selected FPLC fractions representative for main peaks; (**c**) graphical presentation of competitive ELISA of selected FPLC fractions representative for main peaks. Different detection wavelengths for the analysis of elution profile were marked by red (225 nm), pink (297 nm), and blue (280 nm). M, molecular mass marker.

**Figure 3 jcm-10-04499-f003:**
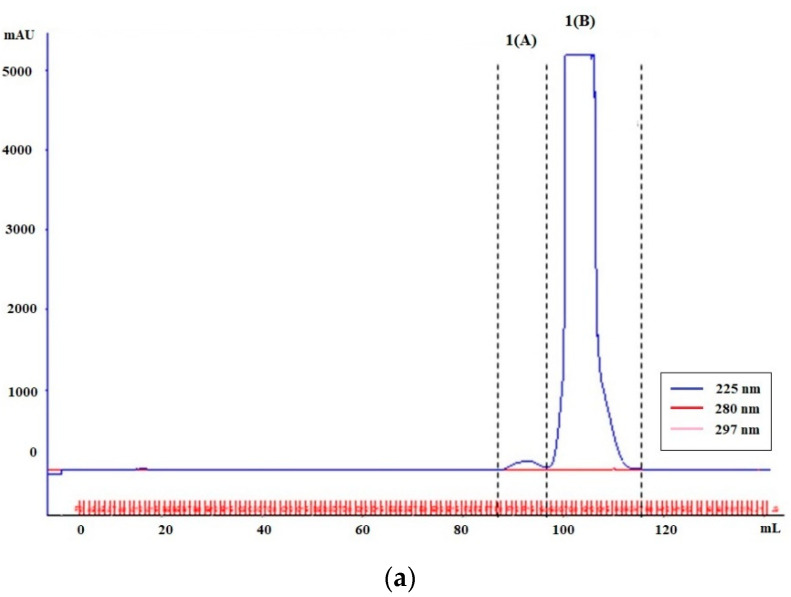
Purification and characterization of low molecular weight glycation products obtained in the synthesis of N-acetyl lysine with melibiose (NAc_α_Lys-MEL) in a microwave reactor: (**a**) The elution profile of NAc_α_Lys-MEL separated by FPLC chromatography (HW40-S column); (**b**) elution profile of “1(B)” peak separated by FPLC chromatography (Bio-Gel P-2 column); (**c**) effectiveness of NAc_α_Lys-MEL fractions 1(A), 1(B0), 1(B1) and 1(B2) in inhibiting MAGE/anti-MAGE reaction; (**d**) mass chromatograms for ion 513.23 *m*/*z* for NAc_α_Lys-MEL peaks 1(A), 1(B0), 1(B1) and 1(B2).

**Figure 4 jcm-10-04499-f004:**
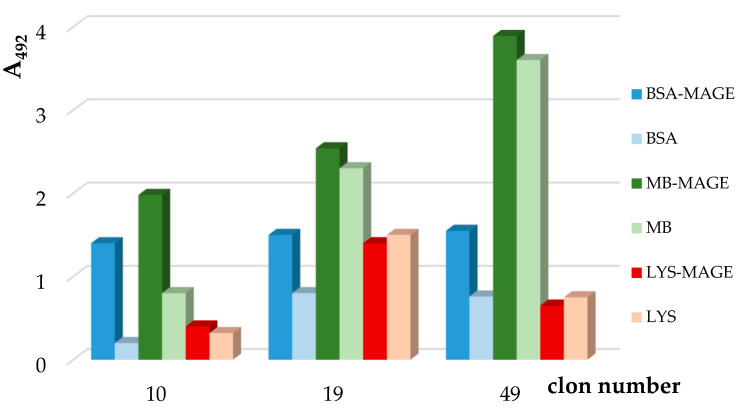
The reactivity of antibodies from Clone Nos. 10, 19 and 49 with BSA-MAGE (bovine serum albumin-MAGE), MB-MAGE (myoglobin-MAGE), LYS-MAGE (lysozyme-MAGE) and with BSA, MB, LYS as respective controls.

**Figure 5 jcm-10-04499-f005:**
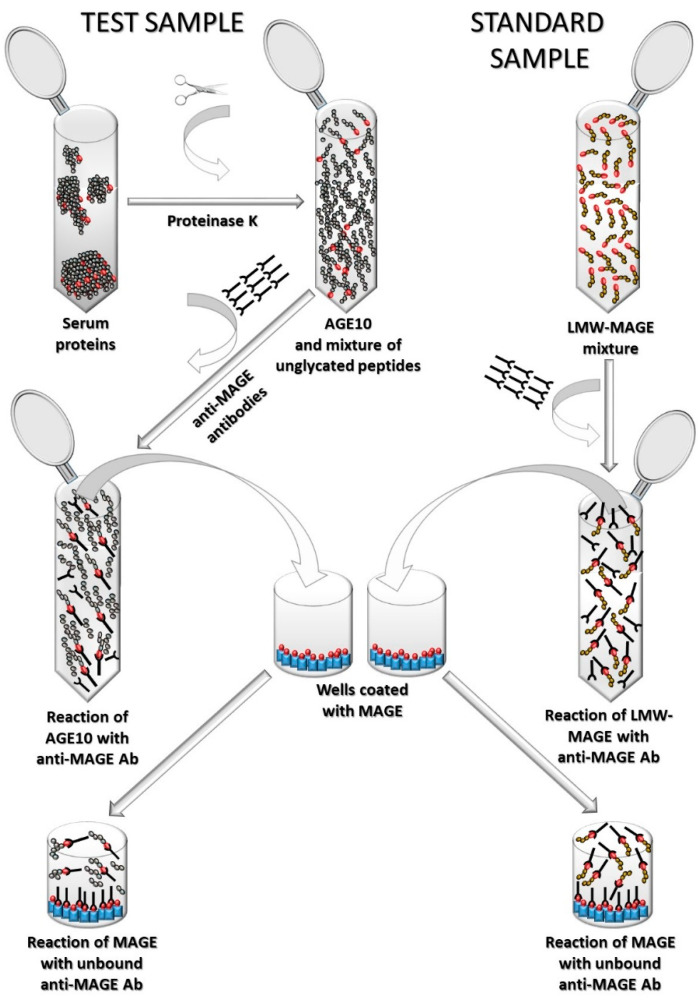
Scheme of AGE10 ELISA. Ab, antibodies; LMW-MAGE, low molecular mass advanced glycation end-products obtained by NAcαLys modification with melibiose (analogue of AGE10 epitope) depicted as red circles; MAGE, advanced glycation end-products obtained by myoglobin modification by melibiose depicted as blue rectangles.

**Figure 6 jcm-10-04499-f006:**
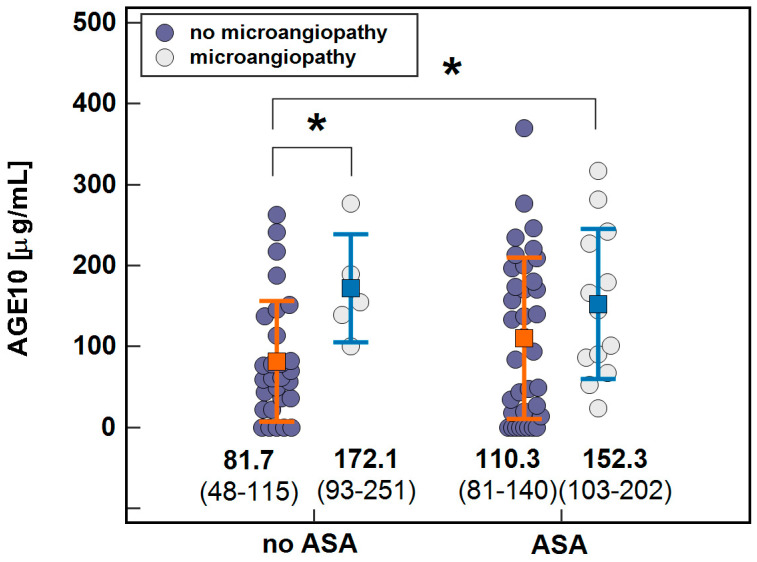
The association of AGE10 with microangiopathy in patients stratified based on aspirin (ASA) treatment. Data presented as means with 95% confidence interval (closed squares with whiskers and figures below the dot-plots) and analyzed using two-way ANOVA. Connectors with asterisks indicate significant (*p* < 0.05) between-group differences.

**Figure 7 jcm-10-04499-f007:**
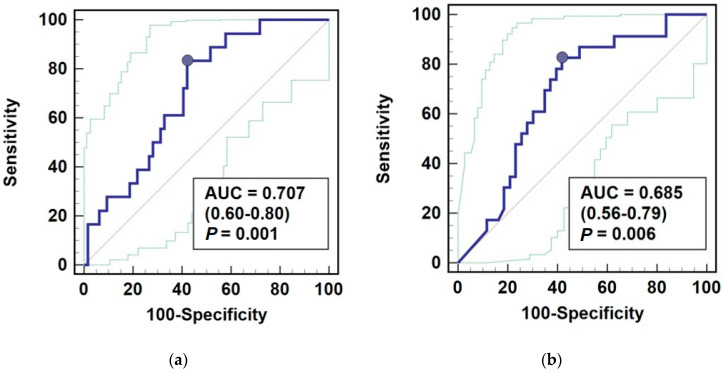
AGE10 as biomarker of: (**a**) microangiopathy; (**b**) stage ≥3 chronic kidney disease. Data presented as receiver operating characteristics (ROC) curves (solid blue line) with 95% confidence interval (dotted green lines). The performance of a chance marker devoid of discriminating power (AUC = 0.5) is represented by diagonal red line and optimal cut-off value is indicated as closed blue circle. AUC, area under ROC curve.

**Table 1 jcm-10-04499-t001:** Characteristics of studied population.

Characteristics	Parameter	Values
Demographics:	Age, mean ± SD	62 ± 10 years
	Sex, *n* [F/M]	51/31
Disease, *n* (%):	T2DM	75 (91.4%)
	T1DM	4 (4.9%)
	T3DM	3 (3.7%)
Complication, *n* (%):	Hypertension	70 (85.4%)
	Hyperlipidemia	67 (81.7%)
	Macroangiopathies	53 (64.6%)
	Microangiopathy	18 (22%)
	CHD	24 (29.3%)
	PAD	21 (25.6%)
	Nephropathy	10 (12.2%)
	Retinopathy	9 (10.6%)
	Polyneuropathy	29 (35.4%)
	Myocardial infarction	9 (10.6%)
	Stroke	9 (10.6%)
Treatment, *n* (%):	Metformin	58 (70.7%)
	Sulfonylurea	35 (42.7%)
	Insulin	33 (40.2%)
	Aspirin	49 (59.8%)
	Clopidogrel	6 (7.3%)
	Oral anticoagulants	10 (12.2%)
Laboratory parameters, mean ± SD:	HbA1c [%]	8.12 ± 2.13
	LDL [mg/dL]	98 ± 34
	HDL [mg/dL]	46.4 ± 15.8
	TC [mg/dL]	180.4 ± 91.2
	TG [mg/dL]	180 ± 91.2
	GLC [mg/dL]	166.8 ± 70.1
	CRP [mg/dL]	2.59 ± 2.18
	Albuminuria [g/dL]	18.76 ± 10.37
	Creatinine [mg/dL]	0.96 ± 0.22
	eGFR [mL/min/1.73 m^2^]	74.11 ± 17.2

*N*, number of observations; SD, standard deviation; F/M, female-to-male ratio; T2DM, type 2 diabetes mellitus; T1DM, type 1 diabetes mellitus; T3DM, other types of diabetes; CHD, ischemic heart disease; PAD, peripheral artery disease; HbA1c, glycated hemoglobin; LDL, low-density lipoprotein; HDL, high-density lipoprotein; TC, total cholesterol; TG, triglycerides; GLC, glucose; CRP, C-reactive protein; eGFR, estimated glomerular filtration rate.

**Table 2 jcm-10-04499-t002:** AGE10 association with complications of diabetes.

Complication	Mean AGE 10 (95%*CI*), *n*		*p*
	Without Complication	With Complication	
Macroangiopathy	110 (76–144), 29	111 (86–137), 53	0.953
Ischemic stroke	108 (86–128), 73	139 (56–223), 9	0.327
PAD	105 (82–128), 61	128 (83–174), 21	0.318
Myocardial infarction	114 (93–136), 73	85 (17–152), 9	0.362
Ischemic heart disease	119 (96–143), 58	90.5 (50–13), 24	0.197
Microangiopathy	98 (75–120), 64	158 (116–200), 18	0.013
Nephropathy	112 (90–134), 72	101 (39–163), 10	0.719
Retinopathy	106 (85–127), 73	152 (83–221), 9	0.156
Polyneuropathy	105 (80–129), 53	123 (86–159), 29	0.404
Hyperlipidemia	81 (46–115), 15	118 (94–141), 67	0.160
Hypertension	95 (51–139), 12	114 (91–136), 70	0.512

*CI*, confidence interval; *n*, number of observations; PAD, peripheral artery disease. Data were analyzed using *t*-test for independent samples.

**Table 3 jcm-10-04499-t003:** Correlation between AGE10 and demographic data, inflammatory, metabolic, and kidney function indices.

Index	Correlation Coefficient (r), *p*
CRP	0.05, *p* = 0.688
Glucose	0, *p* = 0.946
HbA1c	0.04, *p* = 0.735
HDL-cholesterol	−0.15, *p* = 0.218
LDL-cholesterol	−0.05, *p* = 0.684
Triglycerides	−0.11, *p* = 0.360
Creatinine	−0.25, *p* = 0.036
eGFR	0.34, *p* = 0.005
Albumin/Creatinine ratio	−0.21, *p* = 0.171
Age	−0.09, *p* = 0.415

CRP, C-reactive protein; HbA1c, glycated hemoglobin; eGFR, estimated glomerular filtration rate. Data were analyzed using the Pearson test.

**Table 4 jcm-10-04499-t004:** AGE10 association with applied treatment.

Medication	Mean AGE 10 (95%*CI*), *n*		*p*
	Untreated	Treated	
Oral anticoagulants	114 (92–135), 72	92 (29–155), 10	0.495
Aspirin	97 (68–125), 33	120.5 (92–149), 49	0.253
Clopidogrel	106 (86–127), 76	172 (60–283), 6	0.094
Metformin	96 (66–125), 24	117 (91–143), 58	0.340
Sulfonylurea	101 (76–125), 47	125 (90–160), 35	0.242

*CI*, confidence interval; *n*, number of observations. Data were analyzed using *t*-test for independent samples.

## Data Availability

Not applicable.

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
