# Peer review of "Association of Novel Advanced Glycation End-Product (AGE10) with Complications of Diabetes as Measured by Enzyme-Linked Immunosorbent Assay"

_jcm, 2021, doi:10.3390/jcm10194499_

Round 1

Reviewer 1 Report

The present study emphasises the importance of the unique epitope of AGE, the AGE10 which has been identified to play a vital role in DCM complications. the authors have developed an ELISA methodology to identify AGE10 in human serum samples and established a correlation in DCM samples. The manuscript have been well written and explains the methodology in details. 

Major concern: Lysed hemoglobin is a major concern in ELISA dealing with human serum, there is limited data regarding  validation of MB-MAGE antibody for binding specificity. the authors can perform a whole western blot from hPBMC lysate or serum for MB-MAGE specificity. Need data to confirm exclusion of lysed blood/hemaglobin interference with assay.  

Reviewer 2 Report

The accumulation of advanced glycation end-products (AGEs) precedes microangiopathy and the associated diabetic complications, including diabetic nephropathy. The melibiose-AGE epitope (MAGE), or AGE10, has been implicated in patients with autoantibody detection. However, the methodology for quantifications and clinical significance of the AGE10 levels have not been established. In this study, Bronowicka-Szydelko et al. assessed the level of the AGE10 in serum samples from diabetic patients, using a competitive ELISA newly developed in-house with mouse anti-human MAGE monoclonal antibody. The authors found that AGE10 levels were increased with microangiopathy while also positively predicts with kidney function.

Overall, this study represents a novel methodology to assess AGEs levels and the associated microangiopathy complications in diabetic patients. The experimental design, data analysis, and result interpretation are appropriate for the study for the most part. The impact of the manuscript can be further improved by addressing the following concerns:

Major Concerns:

The rationale of using AGE10 as a marker for microangiopathy and chronic kidney disease requires additional discussion regarding its potential physiological function. It is commonly known that diabetic microangiopathy almost always precedes advanced chronic kidney disease. However, the ELISA results presented seemingly dissociate microangiopathy with chronic kidney disease. Unfortunately, the mechanism for such discrepancies remains unclear.

Minor Concerns:

Please correct the text in line 234 on Page 6.

Please optimize figure display to improve clarity and avoid unnecessary graphics (Figure 1-4)

Please correct the tile for Figure 5.

Please reformat Table 2 columns on Page 13
